# Effect of Intensified Training Camp on Psychometric Status, Mood State, and Hematological Markers in Youth Soccer Players

**DOI:** 10.3390/children9121996

**Published:** 2022-12-19

**Authors:** Okba Selmi, Danielle E. Levitt, Ibrahim Ouergui, Bilel Aydi, Anissa Bouassida, Katja Weiss, Beat Knechtle

**Affiliations:** 1High Institute of Sports and Physical Education of Kef, University of Jendouba, El Kef 7100, Tunisia; 2Research Unit, Sportive Sciences, Health and Movement, UR22JS01, University of Jendouba, El Kef 7100, Tunisia; 3High Institute of Sports and Physical Education, Ksar Said, University of Manouba, Tunis 2010, Tunisia; 4Department of Kinesiology & Sport Management, Texas Tech University, Lubbock, TX 79409, USA; 5Institute of Primary Care, University of Zurich, 8006 Zurich, Switzerland; 6Medbase St. Gallen Am Vadianplatz, 9000 St. Gallen, Switzerland

**Keywords:** youth players, mood state, training load, well-being, recovery, blood markers

## Abstract

During training camps, training load is purposefully intensified. Intensified training loads (TL) are associated with psychological variations, increased fatigue, insufficient recovery, decreased muscular performance, and biological changes in adult athletes, but whether these changes occur during training camps in youth athletes has not been established. The aim of this study was to assess changes in psychometric status, vertical jump performance (i.e., height), and hematological markers before and after an intensive training camp in youth soccer players. In this case, 15male youth soccer players (mean ± SD: age: 14.8 ± 0.4 years; height: 172.0 ± 6.9 cm, body mass: 60.8 ± 7.9 kg; training experience: 5.2 ± 0.7 years) completed a 2-week training program consisting of 1 week of moderate TL (MT) and 1 week of intensive training camp (TC). Rate of perceived exertion (RPE), TL, monotony, strain, and psychometric status (total quality of recovery (TQR) and well-being indices (sleep, stress, fatigue, and muscle soreness) were monitored before each first daily training session across two weeks. The profile of mood states (POMS), countermovement jump (CMJ) height, and blood markers (complete blood count, urea, and creatinine) were assessed before and after TC. TL (d = 5.39, large), monotony (d = 3.03, large), strain (d = 4.38, large), and well-being index (d = 7.5, large) scores increased and TQR (d = 4.6, large) decreased during TC. The TC increased tension, fatigue, and total mood disturbance and decreased vigor (all *p* <0.01). CMJ performance *p* < 0.01, d = 0.52, moderate), creatinine (*p* < 0.01, d = 1.29, large), and leukocyte concentration (*p* < 0.01, d = 1.4, large) and granulocyte concentration (*p* < 0.01, d = 1.93, large) increased after TC. Percentage of lymphocytes (*p* < 0.05, d = 1.17, large) and monocytes (*p* < 0.01, d = 1.05, large) decreased while the percentage of granulocytes (*p* < 0.05, d = 0.86, large) increased significantly. Well-being, quality of recovery, mood, granulocyte concentration, and creatinine were all altered during the week-long intensified training camp. These results may provide coaches with valuable information about psychometric status and physiological fatigue and recovery of youth soccer players to better prescribe and adjust training loads during intensive training periods.

## 1. Introduction

Properly planned intensive training phases are necessary to maximize adaptations for competitive performance in soccer. Intensification of training load (TL) during training camp is a common strategy to physically prepare soccer players to cope with the demands of competition [1,2]. During intensified training periods, intensities, volumes, and frequencies increase. Such increases in physical demands are characterized by mental and physical fatigue and have been associated with insufficient recovery [3]. Previous reports in soccer indicated that intensified training camp increased mental and physical fatigue that can be associated with undesirable changes such as inadequate recovery [2,3], resulting in decreased physical performance that may be related to fatigue accumulation [1,4]. Higher TL and persistent fatigue during intensive training periods have been associated with decreased neuromuscular function and explosive strength [1,3,5,6,7]. Other studies have described that high training load and the imbalance between training and recovery during intense training cycles may lead to physical exhaustion, psychometric disturbance, and biological changes that are indicative of nonfunctional overreaching or overtraining syndrome [3,8]. For these reasons, several studies have examined a variety of biological and psychological responses to intensive training periods to identify possible markers of overreaching or overtraining [4,9,10,11,12,13]. Identifying these markers could help coaches to adapt the load and quality of training to maximize adaptations while minimizing negative consequences. Hematological and immune markers change with increasing training load and seem to be sensitive to fatigue and recovery state. For example, Mackinnon [10] reported that heavy TL modified circulating immunoglobulin concentrations, neutrophil and natural killer cell concentrations, and possibly cytotoxic activity. These biological changes illustrate the multisystemic homeostatic perturbations associated with intensified training [6], another possible negative consequence of these necessary training periods.

Markers of psychometric status during TL intensification have received much attention in recent years [2,4,11,14,15]. Subjective methods such as questionnaires that do not require technology to investigate mood, recovery, and well-being in soccer players are preferable because they can easily be used by coaches in the field. Using the well-being indices scale proposed by Hooper and Mackinnon [16], Selmi et al. [17] have shown that one week of intensified training resulted in poorer sleep quality and higher stress, fatigue, and delayed-onset muscle soreness (DOMS) compared with a tapering week in soccer players. Using the total quality of recovery (TQR) scale [18], Selmi et al. [3] reported that TQR significantly decreased in association with decrements in physical performance after two weeks of heavy training in soccer players. Finally, using the profile of mood state (POMS) scale [19], Miranda et al. [20] reported that periods of high TLs resulted in mood disturbances as assessed by the POMS, and increased negative mood scores and decreased vigor score subscales. These changes were associated with hormonal changes and decreased muscular function [20]. On the other hand, during periods of lower TL such as tapering the taper period, these psychometric variables improve [7,21].

Studying the effect of intensive training periods on psychometric status, physical performance, and hematological markers is essential to effectively assess the unintended consequences of intensified TL among soccer players. The majority of work in this area has been performed in adults. In the long-term, sufficient information regarding these parameters in soccer players of different ages, sexes, and competitive levels will allow assessment of individual responses to, and modification of, TLs to maximize adaptations while minimizing the negative consequences. However, no previously published study to date has examined the effect of intensified training during training camp on neuromuscular fatigue, psychometric status, and hematological markers in youth soccer players. Therefore, this study aimed to assess changes in psychometric status, vertical jump performance, and hematological markers before and after an intensive training camp in youth soccer players. 

## 2. Materials and Methods

### 2.1. Participants

In this case, 15 youth soccer players (Mean ± SD: age: 14.8 ± 0.4 years; height: 172.0 ± 6.9 cm, body mass: 60.8 ± 7.9 kg; training experience: 5.2 ± 0.7 years) were voluntarily recruited from the same soccer club. Goalkeepers did not participate in the same training program as players in other positions, so they were excluded from the investigation. All players had a valid sport medical certification and did not receive any medication or consume other drugs during the experimental period. They had no history of injury in the six months prior to the experimentation period, and no injuries were reported during the experimental period. A written parental informed consent was obtained after a detailed explanation of the objectives, procedures, risks, and benefits of participating in the study. The study was approved by the local research ethics committee of the High Institute of Sport and Physical Education of Kef (approval no. 021/2021) and was conducted in accordance with the Declaration of Helsinki.

### 2.2. Procedures

The study was carried out during a 2-week training program [one week of moderate TL (MT) and one week of intensive training camp (TC)]. Before the training program, body weight and height were measured. After measurement of anthropometric characteristics, players performed a graded maximal test (VAMEVAL) [22] to estimate maximal aerobic speed (MAS). This test was also used determine the maximum heart rate (HRmax) for each participant which was subsequently used for prescribing training intensities. The goal of the training camp was to increase TL in to prepare for the physical demands of competition. Technical staff organized and monitored training sessions and TL. During MT, the players performed one training session per day for five days, competed in one official match on the sixth day, and rested the following day. During TC, the players performed ten training sessions across six consecutive days and rested the following day, with two-a-day training (i.e., separate morning and afternoon sessions) on four of the six training days. The total numbers of training sessions, training days, matches, and rest days are shown in Table 1.

After each training session, rating of perceived exertion (RPE) and training duration (min) were collected to calculate TL, monotony, and strain for each player [4]. TQR and well-being indices (subjective sleep, fatigue, stress, and DOMS ratings) were recorded before the first or only training session each day. The profile of mood states (POMS), neuromuscular performance measured by countermovement jump (CMJ) height, and assessment of hematological markers, urea, and creatinine were conducted after the week of moderate training (MT) and after training camp (TC). One week prior to the beginning of the study, players were familiarized with all questionnaires, scales, and the CMJ procedure. The data were collected by the same coach at the same time of day to avoid any possible diurnal variation in the results. All players completed the questionnaires independently and as honestly as possible. All training sessions took place on the same natural grass training field. Details of the training program are present in Table 2.

### 2.3. Training Load Monitoring 

TLs were recorded for every training session during the 2-week study period. TL was calculated by multiplying RPE (reported after each training session) by session (min) and is reported in arbitrary units [23]. The CR-10 scale was used to monitor RPE [23] and was administered 20–30 min after each training session to ensure that reported effort was based on the whole session and not solely the perceived effort at the end of the training session [15]. In addition to training time, the total duration of each session included warm up, cool down, and rest between tasks [15]. When multiple training sessions were performed on a single day, TLs from each session were summed to calculate the day’s total TL. Daily TLs for each week were summed to calculate weekly TL [24]. The ratio between the mean and standard deviation of TLs throughout each week were used to calculate monotony [24]. Weekly TL and monotony were multiplied to calculate strain [24]. These calculations have been previously used to monitor training loads in soccer [3,15].

### 2.4. Well-Being and Recovery State

The well-being indices scale was completed 15 min before the first or only training session each day [2,11,25] and was reflective of the preceding training day response. Each player responded subjectively to questions about their quality of sleep the night prior, stress, fatigue level, and DOMS. Players rated each index using subjective rating scales ranging from 1 to 7 points where 1 indicated “very, very good” (sleep quality) or “very, very low” (fatigue, stress, and DOMS), and 7 indicated “very, very bad” (sleep quality) and “very, very high” (fatigue, stress, and DOMS). The Hooper index (HI) was calculated as the sum of these four indices. Cronbach’s α values for the well-being scales ranged from 0.90 to 0.93. The TQR scale [18] was administered after the well-being indices. The recovery quality scores on the TQR range from 6 (very, very poor) to 20 (very, very good). The scale was completed before the first or only training session each day using the question: “What is your condition now?”. The Cronbach’s α value for the TQR scale was 0.89.

### 2.5. Profile of Mood States (POMS)

The POMS [19] was administrated to monitor mood states. This inventory contains 24 items rated on a 5-point scale from 0 (not at all) to 4 (extremely) in responses to questions about the participants’ current mood. The following six subscales were assessed: tension-anxiety, depression-dejection, anger-hostility, vigor-activity, fatigue-inertia, and confusion-bewilderment. Additionally, a total mood disturbance (TMD) score was calculated by subtracting the score for vigor from the sum of scores for the other five subscales. To prevent a negative score, a constant of 100 was added to the global score [TMD = ((Anger + Confusion + Depression + Fatigue + Tension) − Vigor)) + 100]. The Cronbach’s α for the POMS ranged from 0.85 to 0.93.

### 2.6. Vertical Jump Performance

Vertical jump height was evaluated using the CMJ test which was conducted after performing a standardized 10-min warm-up (6-min run at 65% MAS followed by two sub-maximal CMJs; no passive stretching was allowed) [3]. After the warm-up, participants rested for 3 min before the subsequent test. Each participant started from a standing position, performed a very fast preliminary downward eccentric action followed immediately by a jump with a goal of achieving maximal height. Hands remained on the hips for the entire movement. Jump height was measured using the Optojump system (Microgate SARL, Bolzano, Italy). The subjects performed 2 trials with 1 min of recovery between trials. The best performance was used in the subsequent analysis. 

### 2.7. Blood Sampling and Laboratory Analysis

A medical evaluation was performed by a physician to ensure that the players did not present any sickness that could have impacted the immune profile measurements. Venous blood was collected into EDTA-coated vacuum tubes from the antecubital vein. Blood sampling was performed between 8.00 and 9.00 a.m. after an overnight fast, and participants were at rest for each sample collection. Samples were immediately transported to the laboratory for processing and analysis. Hematological markers were measured in whole blood within 2 h of blood collection using an automated cell counter (Sysmex XN450; Norderstedt, Germany). Analyses included white blood cell concentration, lymphocyte concentration and %, monocyte concentration and %, granulocyte concentration and %, red blood cell concentration, hemoglobin, hematocrit, and platelet concentration. A second vial of blood was centrifuged (3800× *g*, 15 min, 4 °C). Plasma creatinine and urea were measured using a Cobas Integra 400 analyzer (Hoffmann-La Roche, Basel, Switzerland).

### 2.8. Statistical Analyses

The results are expressed as mean ± standard deviation (SD) unless otherwise noted. Percent differences were calculated for each participant and then averaged to provide a mean percent change from MT to TC for each outcome measure. Statistical analyses were performed using SPSS (version 20.0, SPSS Inc., Chicago, IL, USA). Kolmogorov–Smirnov test was used to verify that data fit the assumption of normality. Monotony and strain were calculated for each week. Training loads, well-being indices, and TQR scores were averaged across each week, and the differences between weeks were analyzed using a paired t test. Paired t tests were used to examine differences between measurements after MT and after TC for CMJ performance, POMS scores, and blood markers. Cohen’s d effect sizes were used to interpret the magnitude of the differences between training weeks [26]. The effect sizes were interpreted as follows: trivial: d ≤ 0.20; small: 0.20 < d ≤ 0.50; medium: 0.50 < d ≤ 0.80; and large: d > 0.80 [27]. Statistical significance was set at *p* < 0.05 (power ≥ 0.80). Using these criteria, G*Power software (version 3.1.9.4, Dusseldorf, Germany) was used to determine that a sample size of 15 was sufficient to detect statistically significant differences in our planned outcomes. 

## 3. Results

The analyses included data from 240 individual sessions or matches conducted during the 2-week experimental period. Programming details for each session are in Table 2. Characteristics of each training session are shown in Table 3. TL, monotony, and strain increased during TC compared to MT (TL: +56.7%, d = 5.39; monotony: +33.6%, d = 3.03; strain: +109.5%, d = 4.38; all *p* < 0.001) (Figure 1).

For psychometric status, each well-being index score, HI, and TQR score increased significantly from MT to TC (all *p* < 0.001; Table 4). For mood state, significant changes from MT to TC were observed for the following POMS subscale scores: tension, fatigue, and TMD (all increased *p* < 0.01) and vigor (decreased *p* < 0.01) (Table 5). No significant changes were observed for depression, anger, or confusion (all *p* > 0.05) (Table 5). 

Mean CMJ performance decreased significantly from MT to TC (−5.2%, *p* < 0.001, d = 0.52, moderate; Figure 2). 

Peripheral white blood cell and peripheral granulocyte numbers increased significantly from MT to TC (both *p* < 0.01, d = 1.40 and 1.93 (large), respectively) with no significant change in lymphocyte and monocyte numbers (*p* > 0.05). Significant decreases in percentages of lymphocytes (*p* < 0.05, d = 1.17, large) and monocytes (*p* < 0.01, d = 1.05, large), and a significant increase in the percentage of granulocytes (*p* < 0.05, d = 0.86, large) was observed. No significant differences were observed from MT to TC for red blood cells, hemoglobin, or hematocrit (all, *p* > 0.05). Creatinine significantly increased from MT to TC (*p* < 0.01, d = 1.29, large). No significant difference in urea was observed from MT to TC (*p* > 0.05) (Table 6).

## 4. Discussion

The present study examined well-being, recovery quality, mood, neuromuscular fatigue, and hematological markers during intensive training camp compared to a moderate training period in youth soccer players (U 15). The results showed that increased TL was accompanied by increased monotony and strain and poorer well-being and recovery during the camp. Furthermore, a significant change in mood state (i.e., increased mood disturbance, tension, and fatigue and decreased vigor) was detected after training camp. Finally, circulating immune cell numbers and percentages were altered and creatinine increased after training camp. These parameters are indicative of an overall negative state of recovery and psychometric status associated with worse neuromuscular performance after training camp.

The present study showed that TL increased during training camp, resulting from longer-duration training sessions and increased number and intensity of sessions, leading to a significant increase of monotony and strain. The weekly TL during camp was greater than 2800 (AU), monotony was greater than 2 (AU) indicating monotonous training [28], and strain was greater than 7000 (AU) which is considerably greater than that generally reported in soccer during the in-season period [4,8,15]. These high values may contribute to poorer physical freshness and insufficient recovery, indicative of possible overreaching [24,29], by increasing physical demands. Indeed, Selmi et al. [4] reported that high TL was associated with high monotony and strain, poor recovery, and poor physical readiness. Monotonous training can induce negative biological and emotional changes [3,29] which were also observed in the present study. Overall, the results of the present study suggest that measuring monotony and strain during training camp might be important in monitoring fatigue and manipulating TL to achieve functional overreaching in youth soccer players.

In this study, the present study showed that well-being and recovery quality significantly worsened during training camp. These changes were likely due to higher TL and fatigue accumulation [2,11,30]. These results were similar to those reported by several previous studies demonstrating that TL intensification negatively affected recovery state and well-being [2,4,11,17,25]. Moalla etal. [30] showed that sleep quality, stress, fatigue level, and muscle soreness were associated with daily TL in professional soccer players. Moreover, Selmi et al. [3] showed that these well-being variables and recovery quality were related to sleep-wake rhythm disturbance, fatigue accumulation, pain, and inflammation. Therefore, the negative changes in psychometric status after the week of intensified training camp in the present study can likely be explained by the increased workload. These findings suggested that psychometric status measurements before training sessions and matches may be useful to monitor athlete readiness and, when possible, adapt the individual training load.

In the present study, mood states were negatively affected by training camp with decreased vigor scores and increased fatigue and tension scores. The POMS questionnaire is commonly used to assess athletes’ mood state [6,9,31,32,33,34]. During intensive training cycles, players have generally reported negative mood states [6]. The results of the present study demonstrated that mood state was influenced by TL increase during intense training camp among youth soccer players. These results are in line with those of several previous studies that examined the effects of intensive training and higher TL on mental health in individual and team sports [4,6,34,35]. These results agree also with the results of Selmi et al. [4] that investigated the relationship between high TL and mood state. They indicated that negative mood scores (fatigue, depression, anxiety) and total mood disturbance increased, and vigor decreased after an intensified training period in professional adult soccer players. They reported that decreased vigor during training period may be accompanied by decreased energy and altered cognitive performance. Similarly, Beykzade et al. [36] examined changes in mental health during intense pre-season soccer training. Among the mood subscales, fatigue and depression were sensitive to increased training load. Moreover, female soccer players reported significantly more tension, depression, anger, fatigue, confusion, and total mood disturbance after 7 weeks of intensive training [37]. In the present study, the lack of change in the depression subscale may have been due to the age of the participants (youth compared to adults in previous studies), or the relatively short duration of the intense training camp (one week). However, coaches should be aware that even a single week of intensified training (e.g., training camp) can result in substantial mood disturbances even among youth athletes. 

A notable finding in the present study is the significant decrease in jump height following training camp. It is likely that successive training sessions, persistent fatigue, and insufficient recovery during camp negatively affected participants’ neuromuscular status [3]. This result is supported by previous findings demonstrating that countermovement jump performance decreased during a 5-day intensified soccer-specific training camp in competitive adolescent female soccer players [38]. Furthermore, Selmi et al. [3] showed that jump performance was decreased following a 2-week intensified training cycle in professional adult soccer players. Similarly, Gathercole et al. [39] found that CMJ performance decreased with accumulated TL and poor recovery in elite female national-team rugby sevens athletes. Overall, these results suggest that even a relatively brief intensified training period significantly increases neuromuscular fatigue across sexes and age groups. 

In the current study, red blood cell-related parameters (RBC, Hb, Hct) were unchanged after training camp. However, leukocytes changed during the week-long intensified training period where the white blood cell concentration, granulocyte concentration, and granulocyte percentage increased and lymphocyte and monocyte percentages decreased. In contrast, a previously published study reported that neutrophil and monocyte concentrations decreased after an intensive training period in professional adult soccer players [40]. While monocyte concentrations did not change in the present study, granulocytes showed the opposite result, perhaps due to the age of the athletes or duration of the intensified training period. Granulocytes (i.e., neutrophils) are among the early innate immune responders to muscle damage followed by monocytes which transform into macrophages in skeletal muscle [41]. However, it is unlikely that neutrophils would still be responding to early muscle damage at the time of blood sample collection in this study. Instead, it is possible that the increased neutrophil concentrations in the present study are a compensatory mechanism for decreased neutrophil function that has been observed during intense training periods [10]. The changes in percentage of individual leukocyte subsets in the present study appear to be due to the increased granulocyte concentration without substantial changes in the other subsets. Whether this change in granulocyte concentration after an intense, week-long training camp impacts susceptibility to illness in youth soccer players was not examined in the present study but should be examined in future work. As a precautionary measure, coaches may recommend and emphasize recovery strategies that support immune function (e.g., sleep, hydration, nutrition) for their youth athletes during and after intensified training periods, even single-week training camps.

Collectively, the results from the present study suggest that factors such as increased training session intensity, frequency, and stress could impose a high level of stress on soccer players, leading to alterations in their biological and psychometric demands. This study revealed that a 1-week intensive training camp characterized by increased training load worsened well-being, recovery state, and mood and induced hematological changes among youth soccer players. While these data are important to inform responses to intensified training among youth athletes, there were several limitations. The sample was small and included only male subjects in the U 15 division, limiting the generalizability of conclusions. The training cycle was short, although this provides evidence that even abbreviated intensified training periods can induce changes in the measured parameters. Other physical parameters such as maximal speed should be measured before and after training camp. It would be pertinent to associate the changes observed in this investigation with physical parameters including agility, sprint, endurance, and power performance. Such measures would provide more nuanced evidence regarding performance changes associated with changes in TL, psychometric status, and hematological variables. The absence of anabolic and catabolic hormone measurement is another limitation of the current study because these are useful markers to detect overreaching [3]. Dietary intake was not controlled in this study. Future studies should examine whether nutritional strategies improve responses to intensified training in youth athletes. Finally, relating parameters measured in this study with time-motion variables (i.e., distance covered in a set time, number of sprints, duration of running in different intensity zones, etc.). Since these are important performance metrics, such factors should be considered in future studies.

## 5. Conclusions

In summary, this study demonstrated that increasing TL during training camp negatively affected sleep quality, stress, fatigue, and muscle soreness, recovery, mood, neuromuscular performance, hematological variables, and creatinine among U 15 youth soccer players. The countermovement jump test, psychometric analyses (e.g., TQR scale, POMS questionnaire, and well-being indices), and analysis of circulating immune cells are simple, efficient, and useful methods to monitor fatigue and recovery in youth soccer players. These measures may also be useful for adapting and manipulating TL during intensive training periods in youth soccer players to maximize adaptations while minimizing the risk of overtraining.

## Figures and Tables

**Figure 1 children-09-01996-f001:**
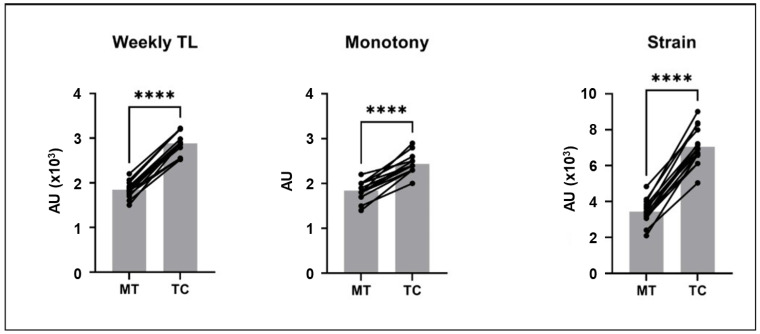
The weekly training load (TL), monotony, and strain during the week of moderate training loads (MT) and the week of intensive training camp (TC), AU: arbitrary units. The gray bars represent means, lines connect values from individual participants from MT to TC. Paired t test, *n* = 15. **** *p*< 0.0001.

**Figure 2 children-09-01996-f002:**
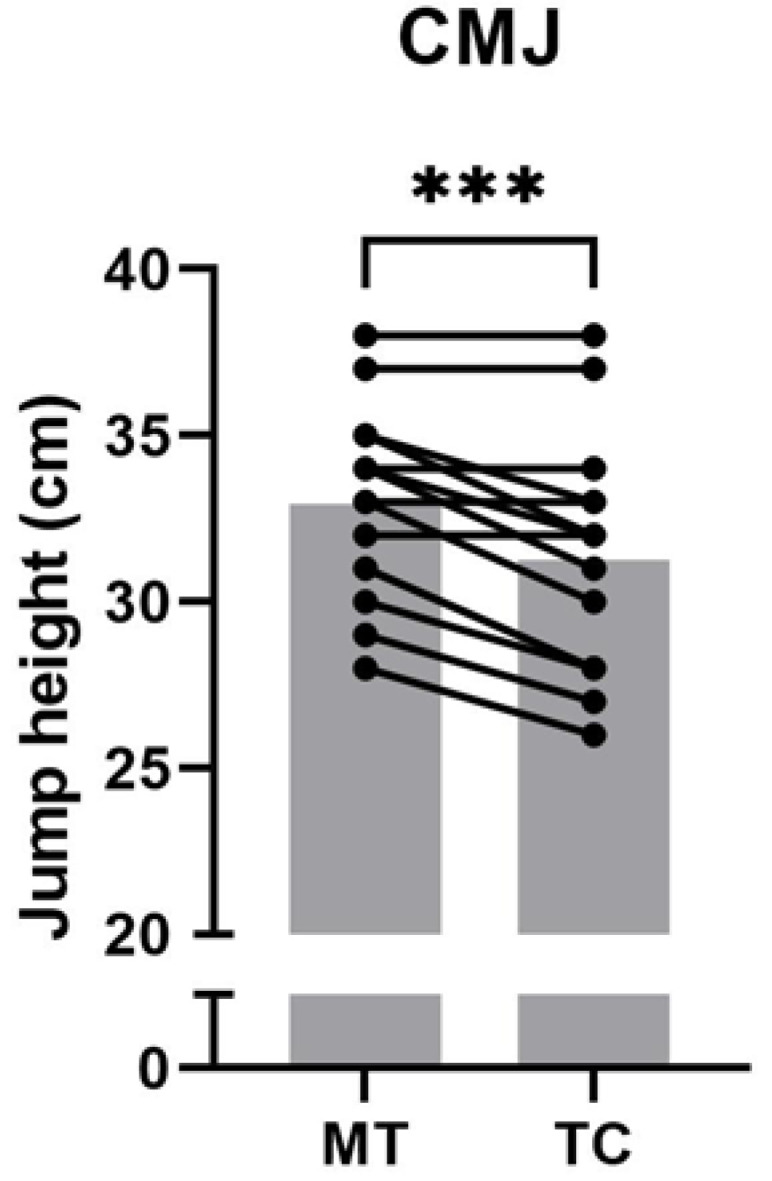
The counter movement jump (CMJ) performance measured after the weeks of moderate training (MT) and intensive training camp (TC). Gray bars represent means, lines connect values from individual participants from MT to TC. Paired t test, *n* = 15, *** *p* < 0.001.

**Table 1 children-09-01996-t001:** The number of training sessions, training days, matches, and rest days throughout the week of moderate training load (MT) and the week of intensive training camp (TC) [4].

	MT	TC
Number of training sessions	5	10
Number of training days	5	6
Number of matches	1	0
Number of rest days	1	1

**Table 2 children-09-01996-t002:** The details of the training program carried out during the moderate training load week (MT) and intensified training camp (TC). MAS: maximal aerobic speed, W = work, R = rest.

		Training Sessions Program		
		Morning/ Afternoon	Physical Program	Technical/Tactical Program
MT	Tuesday	Afternoon	▪ 15 min continuous running (65–70% of MAS). ▪ sheathing exercise (2 sets of 8 exercises: W = 30 s, r = 20 s, R = 2 min)	▪ Passing technique ▪ Possession of the ball on the half pitch
	Wednesday	Afternoon	▪ 20 min proprioception + core stability ▪ 10 min neuromuscular coordination	▪ Doubling exercises ▪ Application of a preferential tactical circuit
	Thursday	Afternoon	▪ Dynamic strengthening with body weight (8 exercises: W = 30 s, R = 20 s). SSG: 3 × 4 min (6 vs. 6 on 35 × 25 m pitch)	▪ Match simulation (2 × 30 min)
	Friday	Afternoon	▪ Agility: 4 exercises [3 repetitions of 15 m each exercise (r = 40 min)], R = 2 min	▪ Tactical work for player positioning
	Saturday	Afternoon	▪ Neuromuscular coordination exercises (based on skipping, mastery of movement and change of direction over short distances) (20 min)	▪ Set ball practice
	Sunday	Afternoon	Match	
	Monday	Rest		
TC	Tuesday	Morning	▪ Continuous running aerobic capacity with ball (Hoff circuit) (2 × 15 min 65–70% of MAS) ▪ Core stability (8 exercises: W = 30 s, R = 20 s)	
		Afternoon		▪ Passing technique: 3 players per group, keeping the ball on a small pitch
	Wednesday	Morning	▪ In gym: General strengthening of all muscle groups: 12 exercises (2 sets of 12 repetitions at each machine (60%1RM), r = 60 s, R = 2 min	
		Afternoon		▪ Passing technique: 6 players per group ▪ Defensive transition
	Thursday	Afternoon	▪ Pyramidal speed drill [2 (5 m, 10 m, 15 m, 20 m; 20 m, 15 m, 10 m, and 5 m), r = 40 s, R = 3 min]	▪ Passing technique ▪ Offensive transition
	Friday	Morning	▪ Dynamic strengthening of the back, abdomen, and core (2 sets of 8 exercises, W = 30 s, r = 30 s, R = 2 min) ▪ High intensity interval training: 3 × 6 min (W = 15 s, r = 15 s at 110 MAS, R = 3 min	
		Afternoon		▪ Passing technique ▪ SSG: 4 vs. 4 + 2 (4 × 4 min, r = 2 min)
	Saturday	Morning	▪ Physical circuit with ball 20 min ▪ Functional training with resistance band ▪ Horizontal and vertical jumping: 4 sets of 4 exercises (4 jumps, r = 30 s), R = 2 min	
		Afternoon		▪ Passing technique ▪ Match simulation on half playing field
	Sunday	Afternoon		▪ Match simulation
	Monday	Rest		

MT: moderate training load, MC: week of intensive training camp, MAS: maximal aerobic speed, W: work, r: recovery between repetition, R: recovery between sets, min: minute, s: second.

**Table 3 children-09-01996-t003:** The training durations and average rating of perceived exertion (RPE), training loads (TL), monotony, and strain during the moderate training load week (MT) and intensified training camp (TC) (*n* = 15).

		Time of Training	Duration (min)	RPE	Session TL(AU)	Daily TL(AU)
Week 1 (MT)	Tuesday	1700	80	3.4	272	272
	Wednesday	1600	90	4.1	369	369
	Thursday	1600	85	3.6	306	306
	Friday	1600	70	3.1	217	217
	Saturday	1600	60	2.2	132	132
	Sunday	1600	90	6.1	549	549
	Monday	Rest	00	0.0	0	0
	Weekly TL(AU)	1845				
	Monotony(AU)	1.84				
	Strain(AU)	3395				
Week 2 (TC)	Tuesday	0900	60	4.2	252	500
		1700	80	3.1	248	
	Wednesday	0900	70	5.0	350	647
		1700	90	3.3	297	
	Thursday	1600	80	3.2	240	256
	Friday	0900	60	4.1	246	502
		1700	80	3.2	256	
	Saturday	0900	75	3.8	285	663
		1700	90	4.2	378	
	Sunday	1600	80	3.9	312	312
	Monday	Rest	00	0.0	0	0
	Weekly TL(AU)	2880				
	Monotony(AU)	2.4				
	Strain(AU)	7064				

AU: arbitrary unit.

**Table 4 children-09-01996-t004:** The averageratings of sleep, stress, fatigue, DOMS, Hooper index (HI) and total quality of recovery (TQR), throughout the weeks of moderate training (MT) and intensive training camp (TC) (values are mean ± SD; *n* = 15).

	MT	TC	Mean % Change	Effect Size (d)	Rating
Sleep	2.4 ± 0.3	3.8 ± 0.3 ***	+59.1	0.4	Small
Stress	2.5 ± 0.1	3.7 ±0.4 ***	+48.3	0.6	Medium
Fatigue	2.6 ± 0.2	3.9 ± 0.3 ***	+50.5	5.7	Large
DOMS	2.1 ± 0.1	3.8 ±0.2 ***	+86.0	10.8	Large
HI	9.5 ± 0.7	15.1 ± 0.8 ***	+58.7	7.5	Large
TQR	15.6 ± 0.9	11.9 ± 0.7 ***	−22.3	4.6	Large

*** *p* < 0.001. HI: Hooper index, DOMS: delayed-onset muscle soreness, TQR: total quality of recovery. Percent changes from MT to TC were calculated for each participant and the average is listed as mean % change. For all measures except TQR, a higher score is negatively associated with well-being.

**Table 5 children-09-01996-t005:** The profile of mood states (POMS) subscale scores for the weeks of moderate training (MT) and intensive training camp (TC) (values are Mean ± SD; *n* = 15).

	MT	TC	Mean % Change	Effect Size (d)	Rating
Tension	3.9 ± 1.1	5.6 ± 1.5 ***	+50.1	1.30	Large
Anger	4.4 ± 1.5	4.4 ± 1.2	+5.9	0.04	Negligible
Confusion	4.4 ± 1.3	4.3 ± 1.4	+2.2	0.13	Negligible
Depression	4.2 ± 1.4	4.3 ± 1.4	+5.5	0.1	Negligible
Fatigue	4.4 ± 1.1	7.8 ± 1.4 ***	+84.0	2.71	Large
Vigor	11.9 ± 1.5	9.5 ± 1.8 ***	−19.0	1.43	Large
TMD	109.4 ± 5.5	116.9 ± 5.5 ***	+6.8	1.35	Large

*** *p*< 0.001, TMD: total mood disturbance. Percent changes from MT to TC were calculated for each participant and the average is listed as mean % change.

**Table 6 children-09-01996-t006:** The change in blood markers from after the week of moderate training (MT) to after intense training camp (TC) (values are mean ± SD; *n* = 15).

	MT	TC	Mean % Change	Effect Size (d)	Rating
WBC (10^3^/µL)	7.27 ± 1.33	9.68 ± 2.15 **	+36.6	1.40	Large
Lym (10^3^/µL)	2.74 ± 0.47	2.63 ± 0.45	+2.7	0.25	Small
Mon (10^3^/µL)	0.73 ± 0.18	0.76 ± 0.11	+8.9	0.21	Small
Gran (10^3^/µL)	3.80 ± 1.36	7.18 ± 2.17 **	+78.4	1.93	Large
Lym (%)	38.59 ± 7.58	28.91 ± 10.44 *	-	1.17	Large
Mon (%)	10.30 ± 1.88	8.88 ± 1.85 *	-	1.05	Large
Gran (%)	51.30 ± 8.60	62.40 ± 13.93 *	-	0.86	Large
RBC (10^3^/µL)	5.34 ± 0.40	5.43 ± 0.43	+1.66	0.22	Small
Hb(g/dl)	13.96 ± 1.1	13.68 ± 1.29	+1.89	0.24	Small
Hct (%)	42.37 ± 2.75	43.81 ± 2.59	-	0.56	Medium
PLT (10^3^/µL)	218.73 ± 37.89	227.27 ± 32.82	+5.0	0.25	Small
Urea (mmol/L)	5.24 ± 1.06	5.25 ± 0.84	+2.9	0.01	Negligible
Creatinine (µmmol/L)	75.66±12.66	87.62±4.93 **	+18.2	1.29	Large

* *p* < 0.05; ** *p* < 0.01. WBC: white blood cells (leukocytes), Lym: lymphocytes, Mon: monocytes, Gran: granulocytes, RBC: red blood cells, Hb; hemoglobin; Hct: hematocrit; PLT: platelets. Percent changes from MT to TC were calculated for each participant and the average is listed as mean % change (except for those outcomes already expressed as percentages).

## Data Availability

The data presented in this study are available on request from the corresponding author.

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
