# Peer review of "Effect of Intensified Training Camp on Psychometric Status, Mood State, and Hematological Markers in Youth Soccer Players"

_children, 2022, doi:10.3390/children9121996_

Round 1
Reviewer 1 Report
First of all, I would like to thank the Editor-in-Chief and Associate Editors from Children journal for giving me the opportunity to have reviewed the Manuscript ID: children-2062881 titled “Effect of intensified training camp in psychometric states mood state and haematological markers in youth soccer players”. The main objective of the present study under review was to assess changes in psychometric status, vertical jump performance, and hematological markers before and after an intensive training camp in youth soccer players. Working hypothesis was that the intensified training loads (TLs) during training camp would change hematological markers and negatively influence well-being, recovery state and mood of youth soccer players. It was a single-club study (in total fifteen outfield youth soccer players participated). No mention was given to the level of the included players. The manuscript seems to have a great degree of originality and may increase knowledge about youth soccer. Main conclusions indicated that perceived well-being, recovery state, mood, immune cell profile, and creatinine were found to be sensitive to fatigue caused by training load intensification. Aside from the prominent novelty of the presented work, I have major revisions that need to be addressed by the authors aiming at improve the manuscript as a whole:
P1L42. “fatigue caused by training load intensification.” - I suggest to have caution to some extent when using the term “fatigue” here as well as across the text.
P2L100. Introduction is presented with various separate paragraphs; combine them into no more than three and avoid extensive literature reviews. Focus on the problem addressed in the present study and the rationale behind the experiment carried out. In addition, the last paragraph should clearly state the importance (both scientific and practical).
P3L115. “Fifteen youth soccer players (Mean±SD: age: 14.8 ± 0.4 years; height: 172.0 ± 6.9 cm, body mass: 60.8 ± 7.9 kg; training experience: 5.2±0.7 years) were voluntarily recruited from the same soccer club.” - It is mandatory to insert sample size calculation to ensure that a sufficient number of participants was reached for the main statistical tests performed latter.
P3L127. “Before the training program, anthropometric characteristics were measured and players performed a graded maximal test (VAMEVAL test) [22] (Selmi et al., 2017) to estimate maximal aerobic speed (MAS) and to determine the maximum heart rate (HRmax), which is subsequently used for prescribing training intensities.” - Cut into two sentences could improve the readability.
P3L139. “I urge the authors to insert as possible more details about the training sessions (e.g. drills prescribed, exercise mode, recovery intervals,…). It can potentially make the study possible to be replicated despite almost not possible as observed in the present form
P3L100. In the Table 1, please insert the reference supporting each of the dependent measures computed.
Table 3 and Table 4. One time the authors stated “MT” and “TC” and another time “T1” and “T2”, please clarify and insert all acronyms in the Table notes to avoid misunderstanding of the readers.
P8L273. If it is difficult to obtain a justifiable number of participants, I suggest to insert post hoc power analysis accompanying p- and d-values.
P4L157. Inclusion of mean and percentage differences together with p-values and effect sizes would increase the practical utility/meaning of the results.
P9L287. The present study examined the well-being, recovery state, mood state, neuromuscular fatigue, and hematological markers during TC in youth soccer players (U 15). The results showed that TL, monotony, and strain increased significantly during the TC. Likewise, well-being indices (i.e., sleep, stress, fatigue, DOMS, and HI) and TQR significantly worsened during TC. Furthermore, a significant change in mood state (i.e., TMD, tension, fatigue increased and vigor decreased) was detected after TC. Finally, circulating immune cells (Numbers and percentages) and creatinine were significantly changed from before to after TC. These parameters are indicative of an overall negative state of recovery and psychometric status associated with worse neuromuscular performance after TC.” There are at least five acronyms in the same sentence, please remove as possible excessive number of acronyms to help improve the flow of the text. If there is no associated maximal number of words as per journal guidelines, I strongly suggest to the authors spell out the full names of things.
P10L352. In general, the discussion is presented with various standalone paragraphs; consider using the following structure to re-write various instances: 1) mention a main finding; 2) compare it with current/past literature, 3) explain (e.g. mechanisms) and make a concluding/recommendation sentence. When making the paragraph of limitations, if the authors find that one or more questions that I pointed out above are not possible to be completed, please discuss as well. It will assist my final decision on the acceptability of the present manuscript for publication. I’m also available to re-review this submission.
Author Response
Reviewer 1
Comment
First of all, I would like to thank the Editor-in-Chief and Associate Editors from Children journal for giving me the opportunity to have reviewed the Manuscript ID: children-2062881 titled “Effect of intensified training camp in psychometric states mood state and haematological markers in youth soccer players”. The main objective of the present study under review was to assess changes in psychometric status, vertical jump performance, and hematological markers before and after an intensive training camp in youth soccer players. Working hypothesis was that the intensified training loads (TLs) during training camp would change hematological markers and negatively influence well-being, recovery state and mood of youth soccer players. It was a singleclub study (in total fifteen outfield youth soccer players participated). No mention was given to the level of the included players. The manuscript seems to have a great degree of originality and may increase knowledge about youth soccer. Main conclusions indicated that perceived well-being, recovery state, mood, immune cell profile, and creatinine were found to be sensitive to fatigue caused by training load intensification. Aside from the prominent novelty of the presented work, I have major revisions that need to be addressed by the authors aiming at improve the manuscript as a whole:
Author’s response:
Thank you for the feedback. We do believe that the paper has improved significantly after your valuable comments that were addressed accordingly.
Comment
P1L42. “fatigue caused by training load intensification.” - I suggest to have caution to some extent when using the term “fatigue” here as well as across the text.
Author’s response:
Thank you for the comment. The term “fatigue” has been examined carefully in each use case and was changed to “training load intensification” in most places.
Comment
P2L100. Introduction is presented with various separate paragraphs; combine them into no more than three and avoid extensive literature reviews. Focus on the problem addressed in the present study and the rationale behind the experiment carried out. In addition, the last paragraph should clearly state the importance (both scientific and practical).
Author’s response:
The various separate paragraphs in the introduction were combined into three paragraphs.
We have minimized the literature review portion of the introduction and included only what was necessary to provide rationale for the present study. Moreover, the scientific and practical relevance was summarized in the final paragraph as follows:
“Studying the effect of intensive training periods on psychometric status, physical performance, and hematological markers is essential to effectively assess the unintended consequences of intensified TL among soccer players. The majority of work in this area has been performed in adults. In the long-term, sufficient information regarding these parameters in soccer players of different ages, sexes, and competitive levels will allow assessment of individual responses to, and modification of, TLs to maximize adaptations while minimizing the negative consequences. To the best of our knowledge, no study to date has examined the effect of intensified TLs during training camp on neuromuscular fatigue, psychometric status, and hematological markers in youth soccer players.”
Comment
P3L115. “Fifteen youth soccer players (Mean±SD: age: 14.8 ± 0.4 years; height: 172.0 ± 6.9 cm, body mass: 60.8 ± 7.9 kg; training experience: 5.2±0.7 years) were voluntarily recruited from the same soccer club.” - It is mandatory to insert sample size calculation to ensure that a sufficient number of participants was reached for the main statistical tests performed latter.
Author’s response:
Thank you for this request. Because no study using this design in male youth soccer athletes has been performed, and we did not have preliminary data from our participants with which to power the study, we conservatively estimated sample size to the best of our ability using previously published work (ours and others). In our most similar previously published study, we reported that after 2 weeks of intensified training (vs 1 week in the present study), professional soccer athletes (adults, vs youth in the present study) had significantly increased training load, monotony, strain, fatigue, DOMS, Hooper index, and worse recovery quality and countermovement jump performance (PMID: 35309535). Effect sizes for these variables were large (d ≥ 0.80). In that study, we also enrolled 15 soccer players, which is the a priori sample size calculated by G*Power software (version 3.1.9.4, Dusseldorf, Germany) to detect a large effect size with an alpha level set at 0.05 and minimum power of 0.80 with a correlation between groups of 0.5 (low for a within-subjects design, but a conservative method of estimation used to power this study). With our anticipated higher correlation between groups (e.g., ~0.90) given the within-subjects design, as few as 10 subjects would be sufficient to detect a moderate effect size (d≥0.50). While we did not measure hematological variables in that study, the inflammatory biomarker C-reactive protein was significantly increased. Regarding hematological variables, neutrophils appear to be leukocyte subset most affected (both in concentration and function) by training intensity (PMID: 11050533).A previous study examining competitive adult distance runners found a significant decrease in neutrophil number during intensified training using only 7 participants in the athlete groups (PMID: 7836192). Therefore, although none of these studies used the same training length, design, or age group as the present study, there was still support that a sample size of 15 would be more than sufficient to detect statistically significant effects in our outcome measures. Indeed, 15 participants was sufficient to detect statistical significance (alpha level = 0.05, minimum power = 0.80) for many of the outcome variables measured in this study.
We performed a post-hoc power analysis using G*Power for the variable with the smallest detected Cohen’s d effect size for which statistically significant differences were observed in the present study (d=0.4, observed for the well-being variable “sleep”). With 15 participants, an alpha level of 0.05, and an observed correlation between groups of 0.73, our analysis indicated that we detected this small effect size with achieved power of >0.99 (screenshot, below).
We have added the following to the end of the “Statistical Analyses" section: “Using these criteria, we used G*Power (version 3.1.9.4, Dusseldorf, Germany) to determine that a sample size of 15 was sufficient to detect statistically significant differences in our planned outcomes.”
Comment
P3L127. “Before the training program, anthropometric characteristics were measured and players performed a graded maximal test (VAMEVAL test) [22] (Selmi et al., 2017) to estimate maximal aerobic speed (MAS) and to determine the maximum heart rate (HRmax), which is subsequently used for prescribing training intensities.” - Cut into two sentences could improve the readability.
Author’s response:
The long sentence was cut into three sentences: “Before the training program, body weight and height were measured. After measurement of anthropometric characteristics, players performed a graded maximal test (VAMEVAL) [22] (Selmi et al., 2017) to estimate maximal aerobic speed (MAS). This test was also used determine the maximum heart rate (HRmax) for each participant which was subsequently used for prescribing training intensities.”
Comment
P3L139. “I urge the authors to insert as possible more details about the training sessions (e.g. drills prescribed, exercise mode, recovery intervals,…). It can potentially make the study possible to be replicated despite almost not possible as observed in the present form .
Author’s response:
More details about the training sessions (e.g. drills prescribed, exercise mode, recovery intervals, etc.) have been added in a new table 2.
Comment
P3L100. In the Table 1, please insert the reference supporting each of the dependent measures computed.
Author’s response:
The reference Selmi et al., 2021 was added [4].
Comment
Table 3 and Table 4. One time the authors stated “MT” and “TC” and another time “T1” and “T2”, please clarify and insert all acronyms in the Table notes to avoid misunderstanding of the readers.
Author’s response:
Thank you for catching this; we apologize for the oversight. “T1” and “T2” were changed to “MT” and “TC” in Table 5 (formerly Table 4). The remainder of the text was also screened to ensure consistency in use of acronyms.
Comment
P8L273. If it is difficult to obtain a justifiable number of participants, I suggest to insert post hoc power analysis accompanying p- and d-values. 
Author’s response:
Thank you for your comment. We have p- and d-values for each of the statistical tests listed/indicated in the tables, and in the graphs, we have bars indicating group means along with lines indicating differences between phases (MT vs TC) for each individual participant in the graphs. Cohen’s d effect sizes are in the text. As the results show, our sample size was sufficient to obtain statistically significant results for most outcomes measured (i.e., TL, monotony, strain, all well-being indices, several POMS subscales and total mood disturbance, CMJ, several hematological variables, and creatinine). Any statistically significant results necessarily had power ≥ 0.80 with an alpha level set at 0.05. We agree that power is important to mention, so we clarified this in the statistical analysis section. The second to last sentence in the “Statistical Analyses” section now reads: “Statistical significance was set at p < 0.05 (power > 0.80).”
Comment
P4L157. Inclusion of mean and percentage differences together with p-values and effect sizes would increase the practical utility/meaning of the results.
We appreciate and agree with your suggestion to add percentage differences to increase practical utility. We have added mean percent changes from MT to TC for each outcome, except for the cell percentages on the hematological analyses (so as not to take a % of a %). The second sentence of the statistical analysis section now reads: “Percent differences were calculated for each participant and then averaged to provide a mean percent change from MT to TC for each outcome measure.”
The original statistical analyses, reported p-values, and Cohen’s d effect sizes are based on means, with each group mean and standard deviation either listed in the respective tables or shown in graphical form. Since each group mean was already listed and we have added in average percent change, we chose not to add the absolute mean difference as an additional column in the tables.
Comment
P9L287. The present study examined the well-being, recovery state, mood state, neuromuscular fatigue, and hematological markers during TC in youth soccer players (U 15). The results showed that TL, monotony, and strain increased significantly during the TC. Likewise, well-being indices (i.e., sleep, stress, fatigue, DOMS, and HI) and TQR significantly worsened during TC. Furthermore, a significant change in mood state (i.e., TMD, tension, fatigue increased and vigor decreased) was detected after TC. Finally, circulating immune cells (Numbers and percentages) and creatinine were significantly changed from before to after TC. These parameters are indicative of an overall negative state of recovery and psychometric status associated with worse neuromuscular performance after TC.” There are at least five acronyms in the same sentence, please remove as possible excessive number of acronyms to help improve the flow of the text. If there is no associated maximal number of words as per journal guidelines, I strongly suggest to the authors spell out the full names of things.
Author’s response:
To improve readability, we removed excessive numbers of acronyms throughout the discussion and opted to spell most things out per your suggestion.
Comment
P10L352. In general, the discussion is presented with various standalone paragraphs; consider using the following structure to re-write various instances: 1) mention a main finding; 2) compare it with current/past literature, 3) explain (e.g. mechanisms) and make a concluding/recommendation sentence. When making the paragraph of limitations, if the authors find that one or more questions that I pointed out above are not possible to be completed, please discuss as well. It will assist my final decision on the acceptability of the present manuscript for publication. I’m also available to re-review this submission.
Author’s response:
Thank you for the comment. We have revised and improved the discussion of the article according to your suggestions.
Reviewer 2 Report
This paper evaluates the changes of some psychological and chemical outcomes and vertical jump performance of young soccer players before and after the intensive training, and tries to explain the negative effects of intensive training on the health, recovery state and mood of young soccer players. This topic has certain significance for the research on the training of young soccer players, and I am very interested in it. However, I still have a few questions and suggestions for this paper.
Q1: As for the materials and methods in the second part, 2.1 participants, I would like to know whether the fifteen participants have never been injured before training? Or haven't been hurt in six months or one year? In addition, it is hoped that the author could clarify whether each subject controlled the diet and other factors that might interfere with the experimental results during the formal training.
Q3. The description of specific intensive training methods in the full text is vague, can you clarify?
Q4. As for the discussion section, I think the influence of fatigue on intensive sports training is significant. The possible mechanism for discussing the results caused by fatigue should not only be a simple description of the test results. In particular, the possible difference in results between adolescent football players and adult athletes as the research object of this study and the potential reasons should be further discussed in the discussion section.
There is a certain amount of literature about the influence of mental and psychological fatigue after intensive training on the athletic performance of football players. The author can further enrich the discussion content after in-depth reading.
Author Response
Reviewer 2
Comment
This paper evaluates the changes of some psychological and chemical outcomes and vertical jump performance of young soccer players before and after the intensive training, and tries to explain the negative effects of intensive training on the health, recovery state and mood of young soccer players. This topic has certain significance for the research on the training of young soccer players, and I am very interested in it. However, I still have a few questions and suggestions for this paper.
Author’s response:
Thank you for the feedback. We do believe that the paper has improved significantly after your valuable comments that were addressed accordingly.
Comment
Q1: As for the materials and methods in the second part, 2.1 participants, I would like to know whether the fifteen participants have never been injured before training? Or haven't been hurt in six months or one year? In addition, it is hoped that the author could clarify whether each subject controlled the diet and other factors that might interfere with the experimental results during the formal training.
Author’s response
Thank you for these questions. All players had a valid sport medical certification and did not receive any medication or consume other drugs during the experimental period. They had no history of injury in the six months prior to the experimentation period, and no injuries were reported during the experimental period. We have added this to the “participants” portion of the methods section. While the training program was controlled and subjects did not consume medications that could have interfered with responses (e.g., NSAIDs), we were unfortunately unable to control all food intake. This has been added to the limitations section.
Comment
Q3. The description of specific intensive training methods in the full text is vague, can you clarify?
Author’s response
Thank you for the comment. A definition of intense training period has been added in the introduction: «During intensified training periods, intensities, volumes, and frequencies increase. Such increases in physical demands are characterized by mental and physical fatigue and have been associated with insufficient recovery [3].”
Details of the training program have been added in a new table (table 2).
Comment
Q4. As for the discussion section, I think the influence of fatigue on intensive sports training is significant. The possible mechanism for discussing the results caused by fatigue should not only be a simple description of the test results. In particular, the possible difference in results between adolescent football players and adult athletes as the research object of this study and the potential reasons should be further discussed in the discussion section.
Author’s response
Thank you for the suggestion; we agree with the importance. We further discussed the results of our study (youth players) with the other studies (adult athletes).
Comment
There is a certain amount of literature about the influence of mental and psychological fatigue after intensive training on the athletic performance of football players. The author can further enrich the discussion content after in-depth reading.
Author’s response
Thank you for the suggestion. After in-depth reading on the influence of mental and psychological fatigue after intensive training in soccer players, the content of the discussion was improved accordingly.
Round 2
Reviewer 1 Report
The current manuscript version contains substantial improvements. The authors carefully addressed my concerns and provided satisfactory answers to all my comments. Based on that, I have recommended publication of this study in its present form.
Reviewer 2 Report
In my opinion, all of the comments raised during the initial review have now been addressed in a satisfactory manner and I have no further concerns or suggestions to report.